# *Opening up Neat New Things*: Exploring Understandings and Experiences of Social and Emotional Learning and Meaningful Physical Education Utilizing Democratic and Reflective Pedagogies

**DOI:** 10.3390/ijerph191811229

**Published:** 2022-09-07

**Authors:** Donal Howley, Ben Dyson, Seunghyun Baek, Judy Fowler, Yanhua Shen

**Affiliations:** 1Department of Kinesiology, Towson University, Towson, MD 21252, USA; 2Department of Kinesiology, University of North Carolina at Greensboro, Greensboro, NC 27412, USA; 3Department of Education, University of Virginia’s College at Wise, Wise, VA 24293, USA

**Keywords:** physical education, physical activity, social and emotional learning, pedagogy

## Abstract

When it comes to teaching social and affective outcomes pertaining to health and physical activity within Physical Education (PE) settings, such learning historically has been observed as manifesting itself as *hoped-for-by-products* rather than intentionally-taught-for curricular outcomes. The purpose of this study was to explore understandings and experiences of Social and Emotional Learning (SEL) and Meaningful Physical Education (MPE) utilizing democratic and reflective pedagogies. A qualitative case study design was implemented in an alternative high school setting in the USA across 10 months. Participants included the Teacher–Researcher (TR), one Physical Education (PE) teacher, a critical friend, two teaching assistants, and 16 ninth-grade alternative high school students aged 14–15 (eight girls/eight boys). Methods involved a TR journal, post–lesson teaching reflections, interviews, and focus groups, with inductive and deductive analysis applied. The following themes were constructed: *It really made you think; making movement meaningful; being a better classmate; and doing things differently*. Results demonstrate how utilizing democratic and reflective approaches grounded in social constructivist learning theory innovatively promoted SEL and MPE. It allowed students to reflect, interrogate and discuss how movement experiences inside and outside of PE influenced their pursuit of a physically active life. Participants articulated experiencing a more inclusive learning experience that challenged the purpose and subject matter of previous PE and physical activity. Teaching for SEL and MPE using common language and terminology around pre–identified and defined competencies, skills, and features drawn from these conceptual frameworks as demonstrated here, can help contribute to more concrete and uniform learning experiences within and across settings. Doing so led participants to demonstrate more holistic and broader understandings of what constituted participation in PE and physical activity, as well as how to promote and participate in meaningful movement and physical activity within and outside of school to promote healthy living. We call for further embedding of democratic and reflective pedagogies in PE teacher education and professional development that provides teachers and students with the opportunity to do so going forward.

## 1. Introduction

The role and place of Social and Emotional Learning (SEL), “the process through which individuals learn and apply a set of social, emotional, behavioral, and character skills required to succeed in schooling, the workplace, relationships, and citizenship” [1] (p. 12), has become a topic of debate and interest in Health and Physical Education (PE) [2]. It is argued that, historically, “personal, social and emotional development is inextricably woven into PE” [3] (p. 6). Learning outcomes related to social and affective domains are recognized as important health and PE related outcomes in secondary school settings [4,5]. However, scholars have argued that “making SEL the primary learning domain for school PE programs is problematic” [6] (p. 5). The educational and political rhetoric surrounding SEL is not matched by a body of empirical research using contemporary theory, framework, and assessment within PE [4]. Instead, discourse persists as to the extent to which holistic health-related and physical activity learning experiences are truly occurring in PE classrooms [7,8,9,10]. Indeed, when it comes to teaching for such outcomes within PE settings, such learning historically has been observed as manifesting itself as *hoped-for-by-products* rather than intentionally-taught-for curricular outcomes [11,12,13,14].

To address this, there have been calls for the “development of PE curricular that are more deeply connected with the lives of students, and which promote well-being” [15] (p. 458). Such an approach has the potential to promote inclusive physical-activity-promoting environments in and through schools. Aligning with the viewpoint that SEL processes should be further integrated into teaching and learning experiences, there is growing evidence of the benefits of supporting teachers and students “in coming to value PE through experiencing meaningfulness (i.e., interpreting an experience as having personal significance) and recognizing ways participation enhances the quality of their lives” [16] (p. 4). A Meaningful Physical Education (MPE) approach has the potential to further target health-related social and emotional outcomes, as well accomplishing established core cognitive and psychomotor outcomes related to PE, promoting lifelong participation in physical activity [17,18,19]. The purpose of this study was to explore both teachers’ and students’ understandings and experiences of SEL and MPE when utilizing democratic and reflective pedagogies. Utilizing constructivist learning theory, conceptual features of MPE, and a systemic framework for SEL, three research questions guided the study: (1) What prior understandings and experiences of SEL and MPE had the teachers and students encountered? (2) How did they understand and experience SEL and MPE within classes? (3) What impact did the democratic and reflective pedagogies have on their combined experiences and understandings of SEL and MPE?

## 2. Theoretical Framework

Similar to a recent study conducted by the authors [18], this study draws on social constructivist learning theory [20], a theoretical framework that seeks to facilitate moving away from instructional learning focusing on isolated skills and drills in PE towards more meaningful and holistic learning connected to, and embodied within, learners’ contextual lives [17,21]. Teaching and learning in such a way requires “opportunities for students to reflect and deliberate and to share and discuss performance related ideas through a sense of shared meaning and a commitment to balancing individual and group needs” [22] (p. 203). It is a suitable theory to promote SEL and MPE as it supports the idea that teaching and learning should be democratic, reflective, and learner-centered, recognizing: (a) deep understanding and multiple connections support transfer to other contexts; (b) prior knowledge and experience enables further learning; (c) learning is an active process of constructing knowledge; and (d) social and cultural environments influence knowledge construction [20]. Dewey recognized education as a social endeavor connected to issues of democratic societies, arguing that “the true center of correlation of the school subjects is…the child’s own social activities” [23] (p. 90) and “failure to take into account the significant social factors means, nonetheless, an absence of mind and a corresponding distortion of emotional life” [24] (p. 91). Within this, a MPE approach is understood as requiring the enactment of the following five features: social interaction; fun; challenge; motor competence; and personally relevant learning [25]. Central to MPE is the utilization of democratic and reflective pedagogies helping learners understand and experience these features when participating in PE and physical activity [19,26]. 

For SEL, we draw on *The Collaborative for Academic, Social, and Emotional Learning’s* (CASEL) Framework for Systemic Social and Emotional Learning [27,28]. This conceptual framework identifies five interrelated sets of health-related cognitive, affective, and behavioral competencies: self-management; self-awareness; social awareness; relationship skills; and responsible decision making [29]. These competencies involve the targeting of specific skills that, when integrated into teaching and learning, can help facilitate accomplishment of broader learning beyond psychomotor and cognitive outcomes. We posited that enacting democratic and reflective pedagogies, tracking teachers and students prior and evolving experiences and understandings, and following up on these, could lead to more explicitly constructed and refined experiences and understandings of SEL and MPE. In doing so, such utilization might contribute to further understandings and applications promoting participation in PE and physical activity focusing on additionally established researched elements such as autonomy, competence, and relatedness [30,31,32]. Regarding autonomy support, the intrinsic motivation required to participate in physical activity might be derived from teachers and students implementing meaningful democratic and reflective learning experiences inside and outside of PE [33,34,35]. This, in turn, could help justify their utilization and inform the refinement of such pedagogies in future work seeking to understand and promote participation in PE, physical, and health-related activities. In the next section, we present the methodological approach which guided this study.

## 3. Methodology

### 3.1. Research Design, Context, and Participants

In line with the ethical procedures approved by the school district and university’s office of research and integrity, participants and the school have been assigned pseudonyms (Study Code: 21-0312). A qualitative case study design was implemented spanning ten months from February to December 2021 [36]. It was conducted as part of a larger study exploring teachers’ and students’ understandings and experiences of SEL and MPE and follows up on an initial participatory-action-research study that took place at the same site involving students from February to May 2021 [18]. The convenience sampling procedure was used for ease of accessibility to participants [37]. The research was conducted in Tyber College, an urban alternative high school with 250 students, operated in partnership by the local school district and a university where the school’s campus was situated. The U.S. Department of Education defines alternative education schools as any “public elementary/secondary school that (a) addresses the needs of students that typically cannot be met in a regular school; (b) provides non-traditional education; (c) serves as an adjunct to a regular school; or (d) falls outside the categories of regular education, special education, or career/technical education” [38]. Teachers and students had just returned to face-to-face learning in January 2021 following COVID-19 restrictions and closures in March 2020. 

Adult participants included one male Teacher–Researcher (TR), two female teaching assistants, and one female PE teacher who taught student participants (See Table 1). A total of 16 ninth-grade students out of 18 from the PE teacher’s class (eight girls/eight boys) aged 14–15 agreed to participate in data collection (See Table 2). Data was not permitted to be collected from the 16 students in the TR’s ninth-grade class as the school district did not permit research to be performed with students the teacher taught and worked with. The PE course for the two classes was purposefully designed with the intention of developing student understandings and applications of SEL and MPE. Central to this was the enactment of democratic and reflective pedagogies during the course promoting student voice, which were the focus of data analysis in the first study (See Table 3). The course itself was implemented by the TR and PE teacher and teaching assistants across 20 and 16 75-min lessons respectively over 10 weeks. During the course, students participated in cooperative activities, taster sessions and curriculum negotiation before selecting and participating in two units of work. The TR’s class selected basketball and pickleball and the PE teacher’s class selected floor hockey with an additional games and sports creation element. Data collected and analyzed from students in the PE teacher’s class utilizing the pedagogies during the course were excluded from this study [18]. The current study sought to further establish trustworthiness and dependability through triangulating additional and longitudinal data from multiple participants and classes during and after the course. It also allowed the authors to follow up on the initial student results to ascertain evolving experiences and understandings of SEL, MPE and relationships with physical activity after the course was over.

### 3.2. Data Collection

This study utilized a range of qualitative methods sequentially implemented across 10 months before, during and after the course itself including a TR journal, individual and post-teaching reflections with a critical friend, interviews and focus groups. The TR journal contained 22 entries averaging 1838 words in length. It involved the TR reflecting from his experiences and thoughts throughout the study, openly juxtaposing his own observations alongside those from the literature, as well as those received from his students, the PE teacher, and the teaching assistants throughout the study. Supplementing this, the TR completed 20 Post Teaching Reflective Analyses [54,55,56]. Following the course, the TR participated in one interview with a critical friend and conducted interviews with the PE teacher and two teaching assistants to further elicit their experiences and understandings of SEL and MPE during the course. These interviews lasted 45–60 min. The 16 students in the PE teacher’s class participated in two rounds of focus group interviews, once immediately after the course in May 2021, and again in December 2021, lasting approximately 30–45 min each. During these interviews, students were asked to reflect on their experiences and understandings of SEL and MPE prior to and during the course. Data were then organized for analysis. 

### 3.3. Data Analysis

The TR journal entries [22], post–teaching reflection analyses [20], critical friend interview [1], PE teacher interview [1], teaching assistant interview [1], and student focus group interviews [7] were collected, organized, transcribed, de-identified, and stored safely. The *Miles, Huberman, and Saldana Framework for Qualitative Data Analysis* [57], involving data condensation, data display, and drawing and verifying conclusions, was initially implemented. This involved both deductive and inductive combination of comparative and thematic analysis or abduction. While drawing on theoretical and conceptual frameworks to deductively, openly and axially code data, we also relied on inductive reasoning to seek out generative patterns in order to establish a thematic structure for the results [58,59]. Open coding was utilized to identify salient ideas related to topics participants had discussed [60]. The TR read and re-read transcripts and analyzed data using open coding to identify and categorize codes and subsequently their sub-properties and category dimensions based on the degree to which they occurred, refining these salient ideas. Overarching codes were then broken down into coded properties. Open coding was repeatedly implemented throughout the data analysis as new codes and categories emerged and older ones were recoded. In vivo codes were then analyzed and selected to pursue meaning and present data that appropriately represented participants’ experiences, rather than those that neatly lined up with the literature. Descriptive codes were used to identify and group interesting statements or events. To apply scholarly interpretation in developing themes, comparative analysis was implemented across domains. This involved comparing incident against incident among participants for similarities and differences, with incidents found to be conceptually similar to previously coded incidents given the same conceptual label and gradually elaborated and brought into variation [61]. Axial coding was then applied to contextualize the data through conceptualizing, defining categories, and developing categories fully in terms of their properties and dimensions and to account for variation [61]. This helped make connections between different sources of data. 

Before presenting results, we wish to acknowledge that the dependability, credibility, transferability, and confirmability of the data is limited in so far as it relates only to these teachers and students in Tyber College, making generalization difficult to apply. Regarding credibility, we have sought to present an accurate representation through providing a backdrop to the context, a description of participants, the study design, methods, and procedures, methods of recruiting and consent, and how trustworthiness was established. The dependability of the study was also influenced by the ethical compliance required by the local school district. Regarding researcher effect, the TR was not granted permission to collect data directly from students in his class or be present in the PE teacher’s class during school time. This compromised the sample size and limited triangulation of data from his classes solely to his TR journal, post teaching reflections, and the interviews he conducted with the critical friend and the teaching assistants. Additionally, no observations were permitted on the PE teacher and her students by the TR during class and school time, limiting the triangulation of their experiences in this study to the individual interviews, and the focus groups. The TR did not engage or influence the immediate class setting in this regard as requested. The TR and PE teacher had no prior relationship with the teaching assistants or the students prior to the study. The students themselves had not met face to face with each other owing to COVID-19. Regarding prolonged engagement and member checking, the students were interviewed first in May 2021, with a follow up interview in December 2021. In utilizing multiple perspectives, i.e., those of the TR, PE Teacher, teaching assistants, and students, we looked to address potential bias and instead demonstrated a conscious effort to triangulate results that reflected the comments of all participants. A consistent effort was made to search for and discuss elements in the data that did not support or appeared to contradict patterns or explanations that were emerging from data analysis. This was done through discussions with the critical friend and further elaboration on the data by participants involved. The main results are presented thematically, followed by a discussion interpreting significance and meaning in relation to theory and literature. 

## 4. Results

The following themes were constructed as thematic results representing teachers and students’ experiences of promoting SEL and MPE: *It really made you think; making movement meaningful; being a better classmate; and doing things differently*.

### 4.1. It Really Made You Think

Coming into the course, both the teachers and students acknowledged that they had rarely experienced or applied reflective practices in PE or physical activity: “Previously, I would never like ask for a personal biography, or a timeline, or ask them to reflect” (TR, Critical Friend Interview (CFI)); “I really didn’t have a lot of experience and knowledge of embedding reflection” (Myung-Hee, Interview (I)); “It wasn’t something that I had done before” (Melissa, FGF). The opportunity to utilize reflection within and outside of classes allowed teachers to draw on students’ prior knowledge and experiences when planning for teaching and learning: “Getting students to think and reflect in this way helps them to identify moments which have contributed to their current relationship with PE and physical activity” (TR, Journal (J)6); “It was a great opportunity to learn about students…I could understand them more” (Myung-Hee, I). Reflective practices such as the personal biographies, timelines, photovoice, and proclamation of meaningfulness required students to think more deeply about their relationships with PE and physical activity: “It’s allowing social awareness to occur. Students and I are learning more about each other” (TR, CFI); “It made me look back at all I’ve done…it was like the first time someone actually told me to think about it…looking back, I realized I need to calm down, like, take a break, help out some people” (Channing, FGA); “We shared a little bit of what we wrote down with our groups…we were able to learn about our peers” (Aubrey, FGD). Doing so meant reflecting on and recognizing the array of emotions each person experienced: “It was positive and negative because I did have some downs…it was good to reflect back on it” (Melissa, FGA); “If another person had a bad experience, it made it really made you think” (Justin FGD);” “The reflections really opened your mind” (Landon, FGE). 

The regular digital class reflections in particular provided students with the opportunity to reflect on their immediate PE and movement experiences: “They were a way of me expressing myself after what I had done in the PE class” (Cathy, FGB); “The questions required you to do a lot of thinking and responding and applying that to what we were doing” (Aubrey, FGG). Within classes, consistent opportunities for group processing were seen as beneficial for reflection. While such opportunities took time to successfully implement, they became a regular feature of classes: “Allowing students time to reflect and consider their own thoughts, ideas, opinions, and those of others…I think that all worked really well” (TR, Post Teaching Reflection Lesson (PTRL)12); “It was an open space for students to speak… …they could speak more honestly about their feelings and thoughts without the teacher” (Myung-Hee, Interview); “We talked through it, everything we did…I could feel somebody else’s perspective, like if they were or weren’t enjoying it” (AJ, FGC). While the increased integration of reflective practices meant at times less participation in physical activity, it was viewed upon by the TR as essential in promoting a more holistic learning experience: “These are the types of reflections that students need to be provided with in order to make sense of their relationship with PE and PA…each person’s meaning is different. (TR, J4); “By taking a step back from physical activity we’re opening up much more discussion about what it is we’re doing, why we’re doing it, and how we can do it in a way that makes it more meaningful for students” (TR, PTRL12). 

### 4.2. Making Movement Meaningful

Teachers continuously reconsidered how they themselves understood, planned for, and presented meaningful movement experiences to students: “We all have different understandings of meaningfulness…should we be looking at activities and movements more differently and getting a sense from students themselves as to how they experience it and how they want to perform the movements?” (TR, CFI); “What is meaningful to them in physical activity or PE? These questions are something we miss in our daily lives, we really don’t have time to think about that…I really tried to communicate with my students about this” (Myung-Hee, I). Ultimately, the experience led teachers and students to gain a greater understanding and appreciation of what MPE and physical activity encompassed, and how it manifested itself within their lessons and broader lives: “A meaningful movement experience comes from understanding such moments and acknowledging and appreciating them for what they are and involve. They become worthwhile by allowing students develop a sense of what exactly it is that is meaningful about them” (TR, J19); “Finding your strengths in an activity…finding something that amplifies them really helps me love physical activity and learn to like it” (Cathy, FGF); “It’s just the little things that I do that really matters…doesn’t always have to be a sport, but like a little game or activity to me can be meaningful” (Melissa, FGF). The following subthemes articulate teachers’ and students’ experiences and understandings of each MPE feature (i.e., fun, challenge, motor competence, and personally relevant learning) as they were observed and described during the study.

#### 4.2.1. Fun

Students alluded frequently to fun as an overarching feature of PE: “It was really fun when we got to come together and play each other’s games because you can see how creative other people can be” (Sarah, FGD); You need to have like a positive attitude, positive environment, people having fun” (Jack, FGD). Facilitating fun as a primary feature of PE served to promote and integrate other features within learning experiences and then further. Fun and social interaction were regularly talked about by students: “I feel like when people meet new people, that’s when the fun really starts to happen…putting your two heads together…understanding what you like and stuff…you can really feel the enjoyment” (Landon, FGA); “I feel that people were having fun and you could definitely see how we were able to interact with each other” (Auria, FGB); “We wanted to have fun and we wanted to play with our friends” (Aubrey, FGD). 

#### 4.2.2. Social Interaction 

Utilizing Cooperative Learning elements such as face–to–face interaction, positive interdependence and interpersonal and social skills within structured tasks allowed teachers to place a deliberate emphasis on social interaction: “We tried to help facilitate conversations about what was going on in class…what was going on between students” (Teaching Assistant (TA)1, I); “We would create a sense of positive interdependence…they had to have face to face interaction and work on their interpersonal and social skills” (TR, CFI). This, in turn, helped students to interact, communicate and work together while learning: “(The teacher) did a great job helping us meet new people…got us to talk more, interact with each other” (Melissa, FGA); “We could all learn about each other…it was easier to interact and perform together” (Jack, FGD).

#### 4.2.3. Challenge

Teachers consciously planned for differentiation, encouraging students to identify and implement appropriate levels of challenge within tasks: “I really wanted to make them feel some kind of challenge…think about what are some strategies to feel more challenged” (Myung-Hee, Interview); “How do I make sure that this is still equitable, and everybody is accounted for, and everyone feels that they are being challenged?” (TR, PTRL3). This allowed students to embrace challenge in diverse ways together: “Some people who are physically active, it came easy to them rather than people who weren’t…it was hard for them, but I feel like they like everyone was challenged” (Barry, FGC); “(The PE Teacher) modified a lot of the games to make it easier for people who hadn’t really played before. And it was also not too easy for the people who had played the sport and had experience with it” (AJ, FGC); “I learned how I can get out of my comfort zone and participate in things that I’ve never really done before” (Auria, FGF). Doing so, teachers and students learned to have a broader view of challenge beyond interpersonal competition: “I remind students that it’s not about being competitive but setting appropriate challenges for themselves and those around them (TR, J11L9); “A lot of people in class are really competitive…it’s hard for me to back off sometimes” (Melissa, FGA); “Wanting to be the best and knowing you’re the best kind of gives you a better ego, but it can good and bad” (Cathy, FGF).

#### 4.2.4. Motor Competence

Students repeatedly described experiencing and observing improved motor competence: “I never knew what motor competence was…I’ve never ever talked about it in any PE class…we’re starting to learn these new things and applying it to what we’re doing (Landon, FGA); “I don’t respond well to not knowing how to do something. I don’t like to do it in front of people until I learn. I was able to come out of my comfort zone and learn…I got better with that” (Cathy, FGB). The need for more motor competent students to responsibly work together and support others who felt less competent was also acknowledged by students, as exampled here by Landon and Jack when reflecting on their social and affective roles in supporting others to demonstrate cognitive and psychomotor learning. 

“I feel like I pushed everybody on my team. Even if you didn’t play sports…I feel like I pushed to everybody on my team to do something like go and hit the ball, go kick it, go use the hockey thing to hit it, you know? Like, I want to do it. So, even if you’ve never played physical activity, like sports or anything like that, I felt like I pushed everybody that was on my team.” (Landon, FGA)

“I was already physically active. I’m a motor competence person. I can drive myself…I think it may have helped us look at how to make the experience for others that maybe aren’t as physically active better for them…the main overall should be improvement throughout the class.”(Jack, FGG) 

#### 4.2.5. Personally Relevant Learning

When it came to personally relevant learning, teachers felt students struggled to articulate and make personal connections between PE and their day-to-day physically active lives: “A lot of students, asked me “What is personal relevance?.’ They really did not have any ideas about it…they were kind of struggling at first” (Myung-Hee, Interview); “We need to work with students to present more relevant content in PE classes that draws on their broader lives” (TR, PTRL20). This was also reflected in students’ comments: “There were students that went for walks with their family and they did not see that as physical activity…they did not see that as being like meaningful or personally relevant” (TR, CFI); “I included a picture I took of a hiking trip and it made me look back and realize that I may not really enjoy hiking, but it’s just nice to be outside and really just see nature” (AJ, FGC). For others, physical and emotional feeling was something that gave them a sense of personal satisfaction: “It’s just the feeling. I don’t know how to pinpoint. It’s just different. I push myself harder I would say for my sport than I would do school” (Melissa FGF); “I feel like it gives me a different feeling; like, the adrenaline rush that comes with it, I don’t know…just makes me hyper…gives me a lot of energy” (Cathy, FGF). 

### 4.3. Being a Better Classmate

Prior to the course, both teachers and students acknowledged shortcomings in their experiences and understandings of SEL in PE: “SEL skills are something that I have not taught very well in my classes” (TR CFI); “I really didn’t have a lot of experience and knowledge about embedding SEL in PE” (Myung-Hee, Interview); “My middle school…they just throw a bunch of balls out, basketballs, footballs, and we just went and played with the people that we knew. We didn’t really make new friends” (Landon, FGA); “We would just sit around the gym and be on our phones and stuff…we would have one test and that would be the pacer test…we didn’t do any activities with each other” (Aubrey, FGD); “Experiences in middle school were way different…teachers really wouldn’t interact with students…you would see like different types of people…they wouldn’t really mix in” (AJ, FGG). Explicitly and intentionally planning, teaching, and assessing for SEL targeting the five competencies using a variety of practices meant teachers were able to do so more deliberately and consistently: “When I do this explicitly, I’m finding myself teaching these while I teach PE. I’m able to target them in my planning and lessons now” (TR, CFI); “I was embedding more and more SEL into my teaching and students realized that they were doing more and more SEL activities” (Myung-Hee, I). The increased focus on accomplishing SEL related outcomes within PE led to a safer and more inclusive environment and was seen by the teachers and students as an essential prerequisite to learning experiences: “We need to do more as teachers to ensure that we are teaching SEL in PE” (TR CFI); “I think SEL should be embedded in daily teaching…for me like it starts from the moment the students come in classroom” (Myung-Hee, I); “I don’t think there was ever too much emotion” (Auria, FGB); “The environment that she had set up…it made it easy to interact and have emotion” (Jack, FGD). The following subthemes articulate teachers and students’ experiences and understandings of each SEL competency (i.e., self-management, self-awareness, social awareness, relationship skills, and responsible decision making) as they were observed and described during the study.

#### 4.3.1. Self–Management

Regarding self–management, teachers and students alluded to having to focus on managing their varying emotions, thoughts, and behaviors in different situations in PE to pursue a physically active life: “I’ve found students able to talk about their frustrations while performing and moving. I love hearing this…it allows them to think about how they can self–manage” (TR J19); “I hate cheaters, that makes me so frustrated…I hate it…but we always ended up having a lot of fun…the annoyance goes away” (Auria, FGB); “I need to keep that physical fitness with me…find myself a challenge…relieve stress and things like that that” (Khalid, FGG). Students spoke about learning how to demonstrate personal and collective agency in classes: “we were allowed to make our own games…really just utilize different skills…she would tell us as a group to decide” (AJ, FGC); “She gave us directions and then asked us to make up other parts” (Leo, FGF). Working in this way meant students were less restricted when performing skills and tasks and were instead repeatedly encouraged to take initiative demonstrating planning and organization skills. 

#### 4.3.2. Self-Awareness

Regular opportunities occurred for students to develop self–awareness through learning to understand their own emotions, thoughts, and values and how these influences their behavior in PE and across contexts: “Their own history, family backgrounds, personal backgrounds I wanted them to understand themselves through sharing those stories” (Myung-Hee, I); “The ability to grow, like in your mental mind…if I start doing something and it is bringing my mental health down, I need to stop if it’s affecting my physical health” (Auria, FGB); “We had to talk about ourselves and I feel that made me going on in the future more confident and less shy to talk” (Sarah, FGE). In this way, students were required to demonstrate honesty to further examine their feelings, values, and thoughts with the aim of promoting self–efficacy.

#### 4.3.3. Social Awareness

Teachers and students recognized the need to understand the perspectives of and empathize with others with different experiences and backgrounds not just in relation to PE and physical activity: “Students are aware of how other students feel about movement…how they may emotionally respond in situations where they’re out of their comfort zone” (TR, CFI); “Understanding people’s individuality…a lot of people really just don’t enjoy activity or sports…to understand a person’s experience, you have to understand them…letting people express themselves…we need to be more like ‘we accept you’…it’s just a more comfortable environment” (Cathy, FGB); “When you knew the people around you, you weren’t as shy or scared to engage” (Sarah, FGD). Participants’ words here demonstrate how they looked to take others’ perspectives, demonstrate empathy and compassion and show concern for the feelings and experiences of others. 

#### 4.3.4. Relationship Skills

Students described numerous instances of developing relationships through the seeking and offering of support and help during classes and described learning to communicate effectively and develop positive relationships with others: “We just all grew like a friendship relationship” (Melissa, FGA); “I liked that she find a way for all of us to know each other and communicate and collaborate with each other” (Khalid, FGB); “We worked on communication skills…getting to know each other through physical activity” (Jack, FGD); “I think making friends was a big part of it. That helped a lot. We just kind of got to know each other” (Cathy FGF). Opportunities for teamwork and collaborative problem solving were noted: “You’re going to have to work with somebody you’re not going to be alone ever…I learned it in PE…it really made me realize like, dang, I’m always going to need somebody” (Landon, FGA); “I would say it was big on teamwork too” (Melissa, FGA). The emphasis on learning and demonstrating relationship skills required students to resolve conflicts constructively and resisting negative social pressure when working with others: “Connecting with other people through PE…I think it’s hard… just helping you be a better classmate” (Jack, FGG); “When you come together as a team it’s difficult…you still have to work with others to accomplish the group” (Khalid, FGG); “It taught us like togetherness and how to work with people doing different things” (Landon, FGE). Students described numerous instances of learning to receive, seek, and offer support and help during classes: “Taking instructive criticism from people…I got better with listening to what people had to say” (Cathy, FGB); “When they saw other people doing it and struggling, then they felt more comfortable and then they started to participate” (Auria, FGB). In this way, students were able to better navigate tasks in PE with differing learning demands and opportunities.

#### 4.3.5. Responsible Decision Making

The explicit and intentional emphasis on SEL meant teachers looked to make caring and constructive choices related to their personal behavior and social interactions with students: “It was a much more inclusive environment” (TR, PTRL18); “I really try to be inclusive when teaching in my class…the important thing is they feel ‘My teacher is trying to care about inclusion and equity, she’s trying to engage everyone in her class’” (Myung-Hee, Interview); “It was pretty inclusive…everyone was willing to work together and they were willing to include each other (TA2, Interview). Students regularly acknowledged the need to be more inclusive within PE, acknowledging the benefits and consequences of their actions for personal, social, and collective wellbeing: “I learned not to judge people. That was very important.” (Khalid, FGB); “I feel like that class was very open and like very inviting for everyone. Like, nobody felt out of place or left out. Everybody was included” (Cathy, FGF). In this way they were able to make more reasoned judgments about their behaviors and actions in class.

### 4.4. Doing Things Differently

For teachers, the explicit and intentional promotion of SEL and MPE represented a shift from their previous approaches to teaching: “I’ve been guilty of teaching things one way only always. This is the way it has to be done” (TR, J11); “This teaching style is kind of different from my previous teaching experience” (Myung-Hee, Interview). For students, the learning experience was positively received and viewed upon as innovative and distinctive from previous learning experiences in PE: “I liked it cause’ it was all different and somehow it just connected” (Melissa, FGA); “I think everyone came at it like open mindedly…it’s like a new way that we found” (Khalid, FGB); “It was different because we were doing like different activities every day and we were engaging in stuff…I was able to actually enjoy physical activity” (Aubrey, FGD); “I really enjoyed it because it’s not something that I usually do…giving an opportunity to try new things…get involved…it just opens up neat new things” (Sarah, FGE). Innovatively drawing on past and present experiences of PE and physical activity allowed both teachers and students to develop an appreciation of the need to do so to develop a better understanding of MPE: “We need to frame PE classes to ensure they are more outward looking than inward obsessed. Why are we here? What is it we do here that helps you live a physically active life outside of here?” (TR, J9); “My past experiences with gym, I didn’t like it…it was more of like, this is what you’re going to do…it wasn’t anything fun for me” (Cathy, FGB); “My previous experience with PE at middle school was we did the same things every week…in this class you actually wanted to come to PE” (Sarah, FGD). Key to this was the enactment of the democratic pedagogies which allowed students to exercise more agency over the selection and implementation of lesson content: “We never had a say before, it was always what the gym teacher wanted…I felt very listened to, I felt like she understood (Auria, FGB); “She encouraged you to show autonomy” (Barry, FGC). Teachers developed a greater appreciation of the need to work more democratically with students: “You need to listen to them. You need to find ways to involve them…give them responsibility over designing the planning or the learning…you have to give them opportunities to critique what’s going on” (TR, CFI); “Empowering them to choose what to do is really important…to think ‘OK, I will do this’…I really wanted to give them more agency in terms of learning” (Myung-Hee, Interview).

In experiencing and embracing these different approaches promoting SEL and MPE, teachers and students came to establish a more holistic understanding concerning the purpose and presentation of PE and physical activity in their lives: “We need to wrap emotion around physical activity in a way that makes it more meaningful and accessible” (TR, J14); “There were definitely students who had a broader sense of understanding of what physical activity could be…understanding that PE didn’t have to be moving so quickly, and you know getting a heartbeat up” (TA1, Interview); “Showing us like what physical activity can look like in different ways…you don’t have to be so one minded on what physical activity is…it can be anything” (Channing, FGE); “I learned that you don’t have to be very physically active to be able to participate and you don’t have to be the strongest or the fastest, you know? You can still participate without being that” (Cathy, FGF). Through increased focus on the application of SEL and MPE within classes, teachers and students repeatedly reflected on and interrogated what the subject matter of PE should really encompass: “PE needs to take a step back from the physical elements and encompass more holistic and broader components…in being physically educated are we just physically competent or competent in pursuing physical activity and movement?” (TR, J9); “I think it should add more time for SEL…how you’re feeling mentally, your individuality, and what you want to do as far as like your goal for physical exercise” (Auria, FGB); “I think it should be like a good mix between like reflecting and giving grade on your performance” (James, FGG). Regarding transfer, the TR emphasized to the need to ensure better transfer of learning pertaining to SEL and MPE beyond PE classes: 

“Just doing it in PE, I still don’t think that is enough…it has to happen across the school…it has to happen at home…we have to try and create some connection with the community…find all these different spaces in the students’ world…try and fill those spaces with opportunities for students to experience SEL and MPE.” (TR, CFI)

We now look to understand and discuss what can be learned from these understandings and experiences of utilizing democratic and reflective pedagogies to promote SEL and MPE.

## 5. Discussion

There is a lack of qualitative research combining teacher and student experiences and understandings of SEL and MPE, aligning with contemporary theoretical and conceptual frameworks at high school level [4,18]. Results from this study demonstrate how the utilization of democratic and reflective pedagogies explicitly and intentionally promoting SEL and MPE helped both teachers and students to better identify and articulate these experiences and understandings within and outside of class. In doing so, they were able to facilitate a more inclusive holistic learning experience and broaden their understandings around the purpose and subject matter of PE and physical activity. From a social constructivist learning perspective, providing students with “a mix of movement and play experiences that trigger instances from which each student can reflect on and analyze their involvement in learning” offered a means through which the teachers could promote “opportunities for students to develop both personally and in terms of their wider class and whole school contribution” [9] (p. 63). Teachers and students acknowledged shortcomings and limitations in their prior experiences and understandings of SEL and MPE. Enacting democratic pedagogies such as the full value contract, taster sessions, and continuous class consultation promoted novel and established meaningful movement experiences [18,47,48,49]. The utilization of reflective pedagogies within and outside of classes allowed teachers and students to draw on and share prior knowledge and experiences of PE and physical activity when planning for teaching and learning [20]. Asking teachers and students to collectively identify and make connections between their individual backgrounds, experiences, and social contexts facilitated the construction of knowledge in a personally meaningful manner [21]. This in turn led to a deliberate and active process of constructing knowledge to address the knowledge and practice gaps between student participation and learning in PE and their day–to–day physically active lives.

Adopting an MPE approach to teaching and learning also represented a new departure for both teachers and students that helped broaden their experiences and understandings of PE, physical activity and health- related movement. Learning how to ascribe meaningfulness to movement experiences required the teachers and students to better understand and articulate the role each feature played and how they overlapped both inside and outside of PE. In doing so they were able to increasingly recognize and appreciate how such a process involved “a complex mix of individual cognitive and affective elements as well as relational, social and cultural dimensions” [16] (p. 6). This led teachers and students to better understand, identify and distinguish between each feature as classes went on, while also recognizing where they overlapped. Beyond recognizing fun as a primary feature, students came to value the need for social interaction when working together in learning tasks. An area of regular contention for students and teachers was the role competition played in class. Moving beyond the previously existing and narrow lens of competition as encompassing challenge, teachers and students increasingly recognized the need to view challenge as involving the setting of “personal process-oriented goals across domains during competitive activities, such as encouraging teammates, limiting erroneous decisions in game play, or focusing on efficient or consistent skill execution” [16] (p. 7). This was also reflected in more competent students’ increasing efforts to facilitate other students in experiencing motor competency. The utilization of MPE features in this study ensured the continued promotion of students’ motor competency and ongoing accomplishment of psychomotor outcomes through “positioning the personal, affective and intrinsic meanings of learners at the core of curriculum development and pedagogical enactment” [62] (p. 119). Through encouraging students to reflect on and consider the “the individual and the contextually-bound nature of a meaningful experience” within and beyond PE, teachers were able to help students make connections to, and better understand, their relationships with PE and physical activity in their broader lives [25]. 

Explicitly and intentionally planning and teaching for SEL using a variety of democratic and reflective practices meant teachers were able to do so more deliberately and consistently. The targeting of the five competencies and associated skills when planning and teaching led to more inclusiveness and equity compared to students’ previous experiences [4,18,28]. Intentional and explicit opportunities promoting agency required students to work together and demonstrate self-management and responsible decision making. Providing opportunities for student reflection on these learning experiences prompted both teachers to modify lesson content and practices to better meet their needs and interests [44]. In doing so, they were able to develop a deeper understanding of their individual and combined experiences, developing a greater sense of self and social awareness amongst the teachers and students of their pre–existing and developing relationships with PE and physical activity. In particular, the ability to establish and maintain healthy and supportive relationships and navigate conflict constructively with and amongst students within PE was regularly alluded to and observed by teachers when planning for and reflecting on their pedagogical approaches within learning experiences. In this regard, the implementation of Cooperative Learning elements and structures provided a platform through which social learning could be promoted within PE specific tasks [42,43].

This study helps further triangulate and affirm the results of the initial participatory action research study [18], while also presenting students emerging understandings and experiences of SEL and MPE in relation to their physically active lives. Results also demonstrate how the TR and PE teacher also observed increased confidence and an improvement in their ability to facilitate a more inclusive holistic learning experience and broaden their own understanding around the purpose and subject matter of PE and physical activity, within and beyond classes and their wider lives. Social constructivist learning theory posits that “learning and learning behavior change are a holistic process in which the learner is actively constructing knowledge and behavior within the cognitive, physical, and social constraints of the environment” [63] (p. 500). By continuously enacting reflection and democratic practices, teachers and students were able to further identify and articulate experiences and understanding of SEL and MPE. This led both teachers and students to call into question the pre–existing subject matter and presentations of PE they had previously encountered [64,65,66]. The skewed emphasis and educational value placed on movement, levels of physical activity, and teaching sports techniques repeatedly as the crux of PE curriculum and practice in prior experiences was evident [10]. The utilization of democratic and reflective pedagogies facilitated a broader holistic experience and understanding of what PE and physical activity could be beyond a narrow “understanding that winning, skill, competitiveness, perseverance, discipline, speed, strength, fitness and aggression are valuable” [67] (p. 217). Similar to recent work by O’Conner et al. [68], viewing PE and physical activity in this way facilitated teachers and students to further consider “teaching and learning opportunities that particular activities and different forms of participation ‘open up’ for PE” (p. 12).

The results of this study have a number of implications. Theoretically speaking, we see how utilizing democratic and reflective approaches grounded in social constructivist learning theory has the potential to promote SEL and MPE. While advocacy for social constructivist approaches is plentiful in educational settings, applying theory to practice requires learning experiences that go beyond simply changing the learner’s behavior in the immediate school setting. Conceptually speaking, researchers and practitioners in PE, general education and physical activity continue to reconsider the teaching and learning of SEL and MPE using various contemporary conceptual educational frameworks and approaches [4,16,69]. To our knowledge, this is the first study at high school level that has sought to utilize together the concepts of SEL [27,28,29] and the features of MPE outlined by Beni et al. [25]. Teaching for SEL and MPE using common language and terminology around pre–identified and defined competencies, skills and features drawn from these conceptual frameworks as demonstrated here can help contribute to more concrete and uniform learning experiences within and across settings. Practically speaking, teachers should look to implement innovative practices such as those utilized in this study to draw on and further construct knowledge which contributes to a more holistic learning experience in PE and physical activity. Future research must continue to address this oversight in examining evidence–based practice and seek to do so through promoting health-related outcomes. Finally, research must look to find more accurate and accessible ways through which SEL and MPE can be assessed in schools, something the field historically has struggled to do [6,7,11]. We hope the pedagogies implemented in this study present innovative ways through which such learning can be accomplished more frequently and comprehensively in future.

## 6. Conclusions

Calls for a shift in emphasis in teaching and learning in PE and physical activity in schools to further align with health-related outcomes and accomplish SEL competencies and skills continues to attract discourse and debate. Justified concern exists that a greater emphasis on SEL might detract from the accomplishment of cognitive and psychomotor outcomes in PE and physical activity in schools [6]. Utilizing the features of MPE within these pedagogies supplemented the accomplishment of cognitive, psychomotor and SEL outcomes related to PE. Their incorporation into cognitive tasks served as catalysts prompting students to reflect on and articulate their experiences and understandings within and beyond PE, leading to a broader appreciation of how movement and physical activity was, currently, and could be further pursued and incorporated to promote healthy and active lifestyles. Notably, doing so led both teachers and students to question the very relevance and benefit of established and prioritized subject matter typically found in PE relative to the movement cultures and lifestyle activities they more commonly pursued and found meaningful. In this regard, this study demonstrates the need to think about *doing things differently* in PE and physical activity in schools to ensure that a more holistic learning experience relevant to students’ prior and ongoing experiences and physically active lifestyles is provided. In closing, we call for further embedding of democratic and reflective pedagogies in PE teacher education and professional development that provides teachers and students with the opportunity to do so going forward in order to better facilitate students in *really thinking, being a better classmate*, and *making movement meaningful*.

## Figures and Tables

**Table 1 ijerph-19-11229-t001:** Adult Participant Details.

Name	Race/Nationality/Gender	Education/Experience
Teacher–Researcher	Caucasian Irish Male	Ba. Sc. & Ma. Sc. in PE PhD Candidate PE Curriculum and Pedagogy (Four Years) University Instructor and Research Assistant (Four Years) High School/Secondary PE Teacher, U.S. & Ireland (10 Years) PE, Physical Activity & Youth Sports Volunteer (15 Years)
PE Teacher Myung-Hee	Asian South Korean Female	BS in PE PhD Candidate PE Curriculum and Pedagogy (Three years) University Research Assistant (Three years) Elementary classroom teacher, South Korea (Four years) High School/Elementary school PE instructor, U.S. (Three years)
Critical Friend	Caucasian New Zealand Male	PhD, Professor of HPE, Education and Kinesiology across seven higher education institutions in New Zealand, Canada, and U.S. Qualitative Researcher Curriculum and Pedagogy (25 Years) Diploma Elementary & Secondary PE, Ba. Sc in Ed., Ma. of Arts
Teaching Assistant A	African American American Female	Graduate Student in Counselling Middle College Teaching Assistant (One Year)
Teaching Assistant B	Caucasian American Female	Graduate Student in Music Middle College Teaching Assistant (One Year)

**Table 2 ijerph-19-11229-t002:** Student Participant Details.

Student Pseudonym	Self-Identified Gender	Self-Identified Race/Ethnicity
Aamira	Female	African American Muslim
AJ	Male	Hispanic
Alisha	Female	Hispanic
Aubrey	Female	African American
Auria	Female	Caucasian
Barry	Male	Caucasian Christian
Channing	Male	African American
Cora	Female	Caucasian African American
Jack	Male	African American
James	Male	Caucasian
Jess	Female	African American Native American
Khalid	Male	African American
Landon	Male	African American
Leo	Male	African American
Melissa	Female	African American
Sarah	Female	Caucasian

**Table 3 ijerph-19-11229-t003:** Enacted Pedagogies Drawn from Literature.

Week(s)	Pedagogy Drawn/Modified from Literature
1–2	Full Value Contract [39]
1–2	Personal Biography [40,41]
1–10	Cooperative Learning and Group Processing [42,43,44]
2–10	Continuous Class Consultation & Negotiation [45,46,47,48,49]
3–4	Timeline [50]
4–7	Taster Sessions [46,49]
7–10	Photovoice Task 1 [50,51]
9–10	Photovoice Task 2 [50,51]
2–10	Digital Reflections [52,53]
10	Overall Digital Reflection [52,53]

## Data Availability

The data presented in this study are available on request from the corresponding author. The data are not publicly available to protect participants’ privacy.

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
