# Peer review of "Opening up Neat New Things: Exploring Understandings and Experiences of Social and Emotional Learning and Meaningful Physical Education Utilizing Democratic and Reflective Pedagogies"

_ijerph, 2022, doi:10.3390/ijerph191811229_

Round 1

Reviewer 1 Report

Title

Overall, the title might spot interest in the reader. However, the current version of the title is too long. Authors could consider shortening the manuscript title.

Abstract

The abstract is very informative and an accurate presentation of the manuscript content. However, Authors could add more information about the study sample. Also, some sentence for the conclusions could be also added the end of abstract.

Introduction

The introduction provides a great overview of the theoretical framework. However, the introduction is very SEL and MPE centered. Authors could also cover other important aspects related to physical education such as autonomy, competence and relatedness support that is crucial to nurture youngsters’ motivation (Ahmadi et al., 2022).

Ahmadi, A., Noetel, M., Parker, P. D., Ryan, R., Ntoumanis, N., Reeve, J., … Lonsdale, C. (2022, February 4). A Classification System for Teachers’ Motivational Behaviours Recommended in Self-Determination Theory Interventions. https://doi.org/10.31234/osf.io/4vrym

Methodology

Study participants are poorly described. Please add more specific information about study sample. Please check Tables and journal guidelines.

I believe Findings should be entitled “Results”?

Discussion

Compared to the results section, the discussion is relatively short. Authors could provide more deeper discussion about the study results compared to other previous studies.

The discussion could end with a paragraph entitled “Strengths, limitations and future research”. Also, Authors could add a paragraph entitled “Practical recommendations” to clearly point out the most important practical recommendations for the reader.

Author Response

We would like to thank you for your correspondence, time, and efforts in reviewing this manuscript. We apologize for the delay in responding, owing to personal circumstances and members the authorship team being out of office during this period of time. We did not wish to rush the revision process, which may in turn have resulted in unnecessary time lost for the reviewers and editorial office. We have responded to your comments below and made subsequent revisions in line with comments from the other reviewer and editorial office. We have attempted to address your comments regarding specific information on sample size, but we would appreciate further clarity if you believe further revision is necessary based on our response and revisions. The following are responses to your individual comments:  

Overall, the title might spot interest in the reader. However, the current version of the title is too long. Authors could consider shortening the manuscript title.

We agree. We have removed the opening quote.

Abstract

The abstract is very informative and an accurate presentation of the manuscript content. However, Authors could add more information about the study sample. Also, some sentence for the conclusions could be also added the end of abstract.

We have added some additional conclusions to the abstract.

Introduction

The introduction provides a great overview of the theoretical framework. However, the introduction is very SEL and MPE centered. Authors could also cover other important aspects related to physical education such as autonomy, competence and relatedness support that is crucial to nurture youngsters’ motivation (Ahmadi et al., 2022).

Ahmadi, A., Noetel, M., Parker, P. D., Ryan, R., Ntoumanis, N., Reeve, J., … Lonsdale, C. (2022, February 4). A Classification System for Teachers’ Motivational Behaviours Recommended in Self-Determination Theory Interventions. https://doi.org/10.31234/osf.io/4vrym

We agree. In preparing this study and manuscript we did not want to over complicate things by drawing in additional theoretical and conceptual aspects related to participation in PE and PA. We acknowledge that these aspects you mention are important aspects that nurture youngsters’ motivation and can be found at different times within this work. We wanted to approach experiences and understandings of PE and PA with an alternative lens that might contribute to further understandings and applications promoting participation in PE and physical activity focusing on additionally established researched elements such as autonomy, competence, and relatedness (Ahmadi et al., 2022; Holt et al., 2019; Vasconcellos et al. 2020). Within the theoretical framework section, we now acknowledge these more established and researched aspects and our intent to try and inform and connect all of these theories and concepts together. Another point worth noting is that the term MPE can be a little bit misleading as it can appear to only tie into PE. We view the concept of MPE as having the potential to be applied and understand as MPA (Meaningful Physical Activity). This is demonstrated by how the study moves beyond the classroom and into students physically active lives outside of PE and school. Also, when reviewing this, please note we were asked by both the other reviewer and editor to paraphrase sections of the theoretical framework for the purpose of concision.

Methodology

Study participants are poorly described. Please add more specific information about study sample. Please check Tables and journal guidelines.

We agree in part that some further clarification of participants and study sample is necessary but would appreciate further clarity on what further specific information about the study sample you would like to see. This was a qualitative case study where participants at one school site were recruited using convenience sampling. 16 out of 18 students in the PE teacher’s class consented to participate in data collection. Owing to ethics, students in the TR’s class (n=16) were not permitted to be recruited. This is a limitation which we acknowledge and now make clearer in the manuscript. At the same time, all adults involved in the PE classes (The TR, PE Teacher, and two TAs) participated. Out of the 18 students that had the opportunity to participate from the PE teacher’s class, 16 consented (Aged 14-15 at the time). Students across the two classes came from similar demographics and diverse ethnicities in the school districts. Beyond providing student participants’ gender, race, and ethnicity we’re not sure what more detail we can provide without revealing too much. Regarding adult participants, we have added in their genders and already provided significant detail about their experience leading into the study.

I believe Findings should be entitled “Results”?

We agree that this is in line with the journal’s instructions for authors and have entitled this section “Results”. Thanks for bringing it to our attention.

Discussion

Compared to the results section, the discussion is relatively short. Authors could provide more deeper discussion about the study results compared to other previous studies.

We agree. We have trimmed down different areas of the results section to allow for better flow for the reader where too many quotes were at times presented. In the discussion we have further considered the implications these findings have compared to previous studies. To the best of our our knowledge, there is a lack of qualitative research combining teacher and student experiences and understandings of SEL and MPE aligning with contemporary theoretical and conceptual frameworks at high school level. That’s what we believe sets this work apart from previous studies. Not only do we affirm the findings of the initial PAR study which focuses solely on student data, but we also further triangulate and demonstrate how teachers and students further developed and brought their experiences and understandings in PE forward in relation to their relationship with PA in their day to day lives beyond PE and school.

The discussion could end with a paragraph entitled “Strengths, limitations and future research”. Also, Authors could add a paragraph entitled “Practical recommendations” to clearly point out the most important practical recommendations for the reader.

We agree, and we have trimmed the results section to allow further flow and space for a paragraph on implications and future recommendations. Regarding limitations, we have expanded upon these at the end of the methodology section before presenting results.

Reviewer 2 Report

IJERPH 1813426

General Comments

Thank you for giving me the opportunity to review the article: “Opening Up Neat New Things: Exploring understandings and Experiences of Social and Emotional Learning and Meaningful Physical Education Utilizing Democratic and Reflective Pedagogies.” This is an important issue and with screen time, sedentary lifestyle, and the rate of obesity in our youth on the increase. Thus, additional tools that encourage physical activity is essential.  The study explores the benefits of self-reflection of students and teachers in regards to physical education with the purpose of developing a more holistic and long-term approach to physical activity.

In its current state, the article does not appear to accurately fit the journal scope. More reference could be made to link the benefits of the pedagogical approach (to physical activity/education) investigated, to health/public health. There were methodological issues that could be made clearer to better reflect the results.

There appeared to be many similarities to a previous publication by this group which can be considered self-plagiarism; extensive paraphrasing is recommended. “Howley, D., Dyson, B., Baek, S., Fowler, J., & Shen, Y. (2021). “This Is Not Gym”: Enacting Student Voice Pedagogies to Promote 750 Social and Emotional are the same as referenced Learning and Meaningful Physical Education. Frontiers in Sports and Active Living, 3, 1-15. 751 https://doi.org/10.3389/fspor.2021.764613”. The student population and Table 1 (Student Participant Details) is exactly the same as Table 2 in the reference above, apart from the order of the participants. The data provided in table 3 is also the same as table 2 in the mentioned reference, apart from the title.

The sentences are sometimes too long and therefore hard to follow. Consider rewriting when sentences run over 3 lines (this is excluding quotes).

A section on Impact and implications of the research would benefit the paper. More information than what is currently provided is required.

Consider adding information about public health in relation to the purpose, and the findings of this study. In its current state this article seems more appropriate for an Education/ Teaching/ Learning Journal or a Sport Psychology type Journal, as the main findings demonstrate effective teaching methods or pedagogical approaches (for PE).

A section on the ‘Limitations of this study’ would be beneficial.

Statements about Ethics Approval (reference) is needed.

Abstract 

-Include duration of study

-It would benefit the abstract to add a line about the implications/impact of this study.

- Ln 11: Review/edit abstract (and main text) for academic scholarship issues where numbers follow numbers, as opposed to numbers following text e.g. where ’16 9th grade’ should be ‘16 ninth grade’ and/or use age range as international readers may not know how old 9th graders are.

Briefly state the benefits of a physical active lifestyle or what are the issues to be address?

Introduction

Provide some more information on the reasons/need to revise the PE curricular in the context of health and current lifestyle of the population of interest. That is to say, the problem needs to be more specifically identified to justify the research.  Considering that this journal has some focus on Health, it would be beneficial to discuss any gaps in the current teaching practises and how it might be associated with disease/health.

Ln 48: remove ‘in’

Theoretical Framework

Line 59- 103: It was good to see that this section was evidence based with sources of information presented in the body of this text. However, most of this section seems self-plagiarised from the author’s previous publication (Dyson et al., 2021). Therefore, extensive paraphrasing is required (understood, that some phrases are in quotation marks and correctly referenced, but this is also exactly the same as in the published article mentioned below). Consider extensive paraphrasing.

Howley, D., Dyson, B., Baek, S., Fowler, J., & Shen, Y. (2021). “This Is Not Gym”: Enacting Student Voice Pedagogies to Promote 750 Social and Emotional are the same as referenced Learning and Meaningful Physical Education. Frontiers in Sports and Active Living, 3, 1-15. 751 https://doi.org/10.3389/fspor.2021.764613.

Methodology

Reading through the entire paper demonstrated that the experimental design was appropriate to test the hypothesis. But, this section did get confusing when it came to the various timings and sequence of events that contributed to the methods (see below).

Some information provided in this section was appropriate for the research undertaken. But, again there was too much similarity (self-plagiarism) to the previous publication (Dyson et al., 2021). The Research Design, Context and Participants, Duration of Study (“The course itself was implemented across ten weeks February to April in sixteen 75–minute lessons, typically delivered twice a week depending on the school calendar”) was exactly the same (apart from 4 lines about adult participants) as the previous publication. Therefore, there is concerns around what was novel about this research project. The reviewer understands that the previous publication might have been a smaller study linked to this one. But, this is not stated in the text of the current study.

Line 127:  some explanation of the definition of ‘an alternative high school’ or reference is needed.

Line 135 “Owing the school district ethics…..” needs clarification as it or rewriting.

Ethics Approval Reference number and ethics statement is missing.

Line 148: “course itself was implemented across 10 weeks” conflicts with Line 158: “This study utilized a range of qualitative methods sequentially implemented across the 10 months”. Clarification is required around the durations of the various components of this study. This could be briefly described in a sequential manner, in the methods section above to avoid confusion. The reviewer was unsure when reading “lasting approximately 30-45 minutes each (line- 171), how this aligned with the “75-minute lessons” (line 148). Reading through the entire paper demonstrated that the experimental design was appropriate to test the hypothesis.

Some brief information on what the PE/Physical Activity lessons entailed would put the reflections (in the Findings) of these sessions into better perspective.

Data Analysis

Line 191: State the version of the ‘In Vivo’ package used.

Could be shortened and written more concisely.

Findings  

Line 228: ‘Myung-Hee’ (and then throughout this section) is different from ‘Mee-Hyung’ in Table 2, be consistent. And is this the PE Teacher’s actual name? If so was consent given to use it?  If not the actual name this needs to be made clear.

Line 256: “the questions required”, please provide the guided questions used in the interviews.

A table with findings of main themes and sub-themes would provide a good overview to this section, which seems otherwise long and ‘wordy’. Then some quotes might be added in the table to reduce the text of the findings.

THE END

Author Response

Reviewer 2

Thank you for giving me the opportunity to review the article: “Opening Up Neat New Things: Exploring understandings and Experiences of Social and Emotional Learning and Meaningful Physical Education Utilizing Democratic and Reflective Pedagogies.” This is an important issue and with screen time, sedentary lifestyle, and the rate of obesity in our youth on the increase. Thus, additional tools that encourage physical activity is essential.  The study explores the benefits of self-reflection of students and teachers in regards to physical education with the purpose of developing a more holistic and long-term approach to physical activity.

We would like to thank you for your correspondence, time, and efforts in reviewing this manuscript. We apologize for the delay in responding, owing to personal circumstances and members the authorship team being out of office during this period of time. We did not wish to rush the revision process, which may in turn have resulted in unnecessary time lost for the reviewers and editorial office. We have responded to your comments below and made subsequent revisions in line with comments from the other reviewer and editorial office. The following are responses to your comments:  

In its current state, the article does not appear to accurately fit the journal scope. More reference could be made to link the benefits of the pedagogical approach (to physical activity/education) investigated, to health/public health. Consider adding information about public health in relation to the purpose, and the findings of this study. In its current state this article seems more appropriate for an Education/ Teaching/ Learning Journal or a Sport Psychology type Journal, as the main findings demonstrate effective teaching methods or pedagogical approaches (for PE).

We acknowledge that this study provides something of an alternative approach to public health and PE, and physical activity in and through schools. This submission was for a IJERPH special issue entitled "Promoting Physical Activity in and through Schools". The call for abstracts invited submissions exploring 1) The role of physical education in promoting physical activity and enhancing young people’s physical activity opportunities and participation; 2) Schools as inclusive physical-activity-promoting environments; and 3) Pedagogical and innovative approaches to the promotion of physical activity in and through schools. On receiving the abstract, the guest editor deemed this manuscript a good fit for the special issue. We acknowledge your comments that more reference to linking the benefits pedagogical approach to health/public health is necessary and have made a concerted effort to do so. We see this study as a starting point in bridging some of the gaps between participation in physical education and physical activity within and beyond schools in young peoples’ lives. IJERPH has recently published studies around pedagogical approaches primarily focusing on physical education settings which have the potential to contribute to broader physical education and public health interventions and discourse (See Fernández-Espínola et al., 2020; Lamb et al., 2021; Shen & Shao, 2022). We feel that this adds something additional and innovative.

Fernández-Espínola, C., Abad Robles, M. T., Collado-Mateo, D., Almagro, B. J., Castillo Viera, E., & Giménez Fuentes-Guerra, F. J. (2020). Effects of Cooperative-Learning Interventions on Physical Education Students’ Intrinsic Motivation: A Systematic Review and Meta-Analysis. International Journal of Environmental Research and Public Health, 17(12), 4451. https://doi.org/10.3390/ijerph17124451

Lamb, C. A., Teraoka, E., Oliver, K. L., & Kirk, D. (2021). Pupils’ Motivational and Emotional Responses to Pedagogies of Affect in Physical Education in Scottish Secondary Schools. International Journal of Environmental Research and Public Health, 18(10), 5183. https://doi.org/10.3390/ijerph18105183

Shen, Y., & Shao, W. (2022). Influence of Hybrid Pedagogical Models on Learning Outcomes in Physical Education: A Systematic Literature Review. International Journal of Environmental Research and Public Health, 19(15), 9673. https://doi.org/10.3390/ijerph19159673

There were methodological issues that could be made clearer to better reflect the results.

We agree. We have sought to provide further clarity around the methodology of this study as well as differentiating it from the recently published study that drew on similar methods and participants which you have rightly identified. 

There appeared to be many similarities to a previous publication by this group which can be considered self-plagiarism; extensive paraphrasing is recommended. “Howley, D., Dyson, B., Baek, S., Fowler, J., & Shen, Y. (2021). “This Is Not Gym”: Enacting Student Voice Pedagogies to Promote 750 Social and Emotional are the same as referenced Learning and Meaningful Physical Education. Frontiers in Sports and Active Living, 3, 1-15. 751 https://doi.org/10.3389/fspor.2021.764613”. The student population and Table 1 (Student Participant Details) is exactly the same as Table 2 in the reference above, apart from the order of the participants. The data provided in table 3 is also the same as table 2 in the mentioned reference, apart from the title.

We agree with this. The context, intervention, and participants are similar. However, the timeline is different. We believe it is important to connect the evolving experiences and understandings of the students in this study to that which was analyzed in the previous publication. Which is why the pseudonyms are the same. This helps readers like you who have read both papers make connections. We acknowledge that we did not make this clear in the original submission. The previous publication adopted a participatory action research design utilizing solely student data that was collected from the PE teacher’s class during the course. 16 of 18 students in the class agreed to participate in the previous study. This was a very rich data base which we believed deserved its own analysis. Students’ personal biographies (16), timelines (16), photovoice task 1 (15), photovoice task 2 (13), digital reflections (71), and overall digital reflections (n=15) were collected as data and analyzed. Additionally, 4 focus groups were conducted with the 16 students. This data was excluded from this study which adopted a more traditional qualitative case study design approach seeking to triangulate both teachers and student experiences of the course and participating in PA thereafter, focusing especially on the TR’s experience of teaching his class and following up on the PE teacher, TA’s and students experiences and evolving understandings of SEL and MPE after the course had finished. In the limitations section both you and the other reviewer have rightly requested more detail on, we explain some of the ethical considerations we had to adhere to which required us to modify the research design. We have paraphrased and revised the methodology section to articulate this more clearly.

The sentences are sometimes too long and therefore hard to follow. Consider rewriting when sentences run over 3 lines (this is excluding quotes).

An endlessly enduring faux pas the first author demonstrates which is correctly critiqued by close colleagues and reviewers. The first author takes full responsibility, apologizes, and endeavors to do better. Collectively, we have revised the paper and broken up sentences throughout the revision for better reading flow.

A section on Impact and implications of the research would benefit the paper. More information than what is currently provided is required.

In line with comments from you and the other reviewer, an additional section on implications has been added to the end of the discussion.

A section on the ‘Limitations of this study’ would be beneficial.

A limitations section in methodology has been added.

Statements about Ethics Approval (reference) is needed.

These have been added.

Abstract 

-Include duration of study

The duration of this study was 10 months, and this has been included in the abstract.

-It would benefit the abstract to add a line about the implications/impact of this study.

Implications have been added to the abstract as well as in an additional paragraph in discussion.

- Ln 11: Review/edit abstract (and main text) for academic scholarship issues where numbers follow numbers, as opposed to numbers following text e.g. where ’16 9th grade’ should be ‘16 ninth grade’ and/or use age range as international readers may not know how old 9th graders are.

Thank you for bringing this to our attention. We have made the appropriate revisions.

Briefly state the benefits of a physical active lifestyle or what are the issues to be address?

We have stated the issue to be addressed in the opening sentence of the abstract. The issue to be addressed is that while learning outcomes related to social and affective domains are recognized as important health related and PE outcomes in secondary school settings, discourse persists as to the extent to which holistic health-related and physical activity learning experiences are truly occurring in PE classrooms. When it comes to teaching for such outcomes within PE settings, such learning historically has been observed as manifesting itself as hoped-for-by-products rather than intentionally-taught-for curricular outcomes (Bailey et al., 2009; Lamb et al., 2021; Teraoka et al., 2021). This study set out to address this through exploring experiences and understandings of implementing social constructivist learning approaches promoting democratic and reflective practices specifically targeting these domains.

Introduction

Provide some more information on the reasons/need to revise the PE curricular in the context of health and current lifestyle of the population of interest. That is to say, the problem needs to be more specifically identified to justify the research.  Considering that this journal has some focus on Health, it would be beneficial to discuss any gaps in the current teaching practices and how it might be associated with disease/health.

In line with your comments on the abstract, we have expanded on the disconnect between health-related outcomes pertaining to social and affective learning and the teaching and learning that tends to take place in PE settings focusing more so on psychomotor and cognitive outcomes. We have drawn on additional literature to try and make more explicit connections (and indeed point out disconnects) between Health, PE, and PA in the introduction.

Ln 48: remove ‘in’

This has been removed.

Theoretical Framework

Line 59- 103: It was good to see that this section was evidence based with sources of information presented in the body of this text. However, most of this section seems self-plagiarised from the author’s previous publication (Dyson et al., 2021). Therefore, extensive paraphrasing is required (understood, that some phrases are in quotation marks and correctly referenced, but this is also exactly the same as in the published article mentioned below). Consider extensive paraphrasing.

Howley, D., Dyson, B., Baek, S., Fowler, J., & Shen, Y. (2021). “This Is Not Gym”: Enacting Student Voice Pedagogies to Promote 750 Social and Emotional are the same as referenced Learning and Meaningful Physical Education. Frontiers in Sports and Active Living, 3, 1-15. 751 https://doi.org/10.3389/fspor.2021.764613.

We’re glad you’ve read our previous article and agree with the theoretical and conceptual underpinnings of our approach to both that and this study. That publication tees this manuscript up nicely. We acknowledge that much of this section was drawn from previous theoretical work and literature conducted by the same authorship group in this article. Both formed part of a larger study, where at times a lot of literature and sections intersected and were repeatedly alluded to. We have permission rights from the previous open access publisher to reproduce material that is ours. However, we agree that it is more productive and proper to cite and paraphrase literature we have previously published. We have done so, while also providing additional literature we have come across to further frame our hypothesis. We have provided additional literature around social constructivist learning theory in the discussion section also.

Methodology

Reading through the entire paper demonstrated that the experimental design was appropriate to test the hypothesis. But, this section did get confusing when it came to the various timings and sequence of events that contributed to the methods (see below).

We agree and we hope our revisions in the methodology section provide further clarity on this.

Some information provided in this section was appropriate for the research undertaken. But, again there was too much similarity (self-plagiarism) to the previous publication (Dyson et al., 2021). The Research Design, Context and Participants, Duration of Study (“The course itself was implemented across ten weeks February to April in sixteen 75–minute lessons, typically delivered twice a week depending on the school calendar”) was exactly the same (apart from 4 lines about adult participants) as the previous publication. Therefore, there is concerns around what was novel about this research project. The reviewer understands that the previous publication might have been a smaller study linked to this one. But, this is not stated in the text of the current study.

We agree that more explicit reference to the previous study and paraphrasing was needed. We have sought to provide further clarity on this in the research design section to differentiate between the two studies while at the same time connecting them within the larger study.

Line 127:  some explanation of the definition of ‘an alternative high school’ or reference is needed.

A referenced definition is provided, drawn from the previous publication.

Line 135 “Owing the school district ethics…..” needs clarification as it or rewriting.

We have rewritten this for further clarity: Data was not permitted to be collected from the 16 students in the TR’s ninth-grade class as the school district did not permit research to be performed with students he taught and worked with on a daily basis.

Ethics Approval Reference number and ethics statement is missing.

We have now provided the ethics number and ethics statement.

Line 148: “course itself was implemented across 10 weeks” conflicts with Line 158: “This study utilized a range of qualitative methods sequentially implemented across the 10 months”. Clarification is required around the durations of the various components of this study. This could be briefly described in a sequential manner, in the methods section above to avoid confusion. The reviewer was unsure when reading “lasting approximately 30-45 minutes each (line- 171), how this aligned with the “75-minute lessons” (line 148). Reading through the entire paper demonstrated that the experimental design was appropriate to test the hypothesis.

We agree that this was confusing, and further clarification was required. We have provided further clarity in the methodology section.

Some brief information on what the PE/Physical Activity lessons entailed would put the reflections (in the Findings) of these sessions into better perspective.

We have provided further information on this: During the course, students participated in cooperative activities, taster sessions, curriculum negotiation before selecting and participating in two units of work. The TR’s class selected basketball and pickleball, the PE teacher’s class selected floor hockey with an additional games and sports creation element.

Data Analysis

Line 191: State the version of the ‘In Vivo’ package used.

The data analysis was conducted manually and not with software. In-Vivo refers to words or terms used by the interviewees that may not have initially been deemed significant but were then deemed contextually relevant and notable that they should also be analyzed as codes. 

Findings  

Line 228: ‘Myung-Hee’ (and then throughout this section) is different from ‘Mee-Hyung’ in Table 2, be consistent. And is this the PE Teacher’s actual name? If so was consent given to use it?  If not the actual name this needs to be made clear.

We now state that participants and the school have been assigned pseudonyms at the beginning of the methodology section. The correct pseudonym for this participant is Myung-Hee. However, this was miswritten in the table. This has been revised.

Line 256: “the questions required”, please provide the guided questions used in the interviews.

The student in this instance is speaking in the December focus group (FGG) about the experience of responding to the digital reflections which they completed after certain lessons during the course.

A table with findings of main themes and sub-themes would provide a good overview to this section, which seems otherwise long and ‘wordy’. Then some quotes might be added in the table to reduce the text of the findings.

We agree the original results section was somewhat drawn out. At the same time, we’re glad we could show you all of these quotes in the review process. We typically present our results in this way, and we feel from a qualitative perspective it allows the reader to access and understand the findings more readily, representing the participants’ experiences and understandings in sequence. In line with reviewer and editor comments, and given that we have had to expand on other sections of the paper, we have revised and cut elements of the results to be more concise and allow for better flow when reading.  

Round 2

Reviewer 1 Report

Authors have done well job on revising the manuscript. I have only one concern left. Specifically, Authors argue that providing autonomy, competence and relatedness support is important to foster students' participation in physical education and their physical activity. This information is correct. Previous research has shown that the autonomy support is the most important factor to enhance adolescents' daily physical activity. In addition, Authors could specify that the mechanism by which autonomy support improves students' physical activity is via their intrinsic motivation (Kalajas-Tilga et al., 2020).

Kalajas-Tilga, H., Koka, A., Hein, V., Tilga, H., & Raudsepp, L. (2020). Motivational processes in physical education and objectively measured physical activity among adolescents. Journal of Sport and Health Science, 9(5), 462–471. https://doi.org/10.1016/j.jshs.2019.06.001

Author Response

Thank you once again for your time and efforts in reviewing this manuscript. We are pleased we have addressed initial comments to your satisfaction, bar one concern. We have now addressed this concern in the theoretical framework section, where we extend on our hypotheses to discuss intrinsic motivation with your suggested literature source and additional sources which also align with the study's theoretical and methodological underpinnings.  

In doing so, such utilization might contribute to further understandings and applications promoting participation in PE and physical activity focusing on additionally established researched elements such as autonomy, competence, and relatedness (30,31,32). Regarding autonomy support, the intrinsic motivation required to participate in physical activity might be derived from teachers and students implementing meaningful democratic and reflective learning experiences inside and outside of PE (33,34,35).  

Author Response

Thank you for your time and effort in reviewing this manuscript. We're glad that you are satisfied with our efforts to respond to your comments through our revisions, which have undoubtedly improved the quality of the manuscript for the journal audience.